# Gut Microbial Signatures of Broiler Lines Divergently Selected for Inosine Monophosphate and Intramuscular Fat Content

**DOI:** 10.3390/ani15162337

**Published:** 2025-08-09

**Authors:** Yaodong Hu, Pengxin Cui, Shunshun Han, Xia Xiong, Qinke Huang, Xiaoyan Song, Guo He, Peng Ren

**Affiliations:** 1College of Animal Science, Xichang University, Xichang 615000, China; xcc20210231@xcc.edu.cn; 2College of Life Sciences and Agri-Forestry, Southwest University of Science and Technology, Mianyang 621010, China; 15138181151@163.com; 3Farm Animal Germplasm Resources and Biotech Breeding Key Laboratory of Sichuan Province, College of Animal Science and Technology, Sichuan Agricultural University, Chengdu 611130, China; hanshunshun@sicau.edu.cn; 4Animal Breeding and Genetics Key Laboratory of Sichuan Province, Sichuan Animal Science Academy, Chengdu 610066, China; xiongxia20120904@163.com (X.X.); babalasxy@163.com (X.S.); 5Guangyuan City Animal Husbandry Seed Management Station, Guangyuan 628107, China; hxm_56557140@163.com; 6Wanyuan Animal Disease Prevention and Control Centre, Wanyuan 636350, China; 18282903029@163.com

**Keywords:** broiler, inosine monophosphate, intramuscular fat, metagenome, gut microbiota, metagenome-assembled genome

## Abstract

Understanding the impact of gut microbiota on the deposition of flavor-related compounds—such as inosine monophosphate (IMP) and intramuscular fat (IMF)—in chicken meat is critically important, yet it remains poorly explored. Therefore, we compared the gut microbiota in two lines of Daheng broilers selectively bred for high and low IMP/IMF levels. Significant differences in the cecal microbial communities were observed between the two groups. In particular, specific bacterial taxa—especially those within the *Phylum bacteroidota* and the species *Merdivivens faecigallinarum*—as well as microbial functions related to energy metabolism and nutrient utilization were enriched in the high-IMP/IMF groups. Furthermore, we constructed a comprehensive catalog of chicken gut bacterial genomes based on metagenome-assembled genomes (MAGs). This study provides new insights for improving meat quality in commercial poultry breeding.

## 1. Introduction

With the sustained improvement in living standards over recent years, poultry meat consumption has continued to rise, while consumer demand for the sensory quality of meat has also increased significantly [1]. Inosine monophosphate (IMP) and intramuscular fat (IMF) are pivotal biomarkers of broiler meat quality [2,3,4]. The umami-enhancing effect of IMP is closely associated with the hydroxyl, amino, and phosphate groups of its purine ring, and its synergistic interaction with glutamate can substantially intensify umami perception [5,6]. Phospholipids and polyunsaturated fatty acids derived from IMF contribute to the formation of volatile flavor compounds during thermal processing [7]. Despite their importance, commercial broiler strains optimized for rapid growth often exhibit reduced IMP/IMF deposition due to metabolic trade-offs [8], highlighting the need for novel strategies to improve meat quality.

The genetic regulators of IMP/IMF have been extensively documented. IMP accumulation is modulated by candidate genes, including *PKM2*, *PGM1*, *TGIF1*, *THBS1*, and *AMPD1*, as well as multiple long non-coding RNAs [9,10,11,12]. In contrast, IMF deposition involves the transcriptional control of adipogenesis, mediated by genes such as *CPT1A*, *COL6A1*, and *FASN* [13,14,15]. A substantial body of research has demonstrated that host meat quality is significantly influenced by various factors including breed, nutrition, rearing practices, and developmental stage [6,16,17]. Notably, female chickens possess stronger lipogenic capacity compared to males, and estrogen may further enhance fat synthesis. As a result, hens tend to accumulate higher levels of IMF, leading to improved meat tenderness, lower shear force, and better flavor quality, which are more aligned with consumer preferences [18]. However, compared to these host-related factors, the regulatory role and underlying mechanisms of the gut microbiome in modulating IMP/IMF deposition remain poorly understood, particularly in genetically specialized breeding lines.

To strengthen the rationale for using IMF and IMP as models for host–microbiota interaction studies, recent research on divergently selected broiler lines provides valuable insights. For instance, a long-term divergent selection for abdominal fat in broilers revealed significant differences in gut microbiota composition and function, with fat and lean lines showing distinct microbial taxa and enriched pathways related to lipid metabolism, such as the citrate cycle and PPAR signaling, indicating a co-evolution between host genetics and microbiota that affects fat deposition phenotypes [19]. Similarly, multi-omics analyses of obese and lean broiler lines demonstrated strong correlations between specific intestinal microbes (e.g., *Alistipes*, *Ruminococcaceae*) and host gene expression involved in lipid metabolism and immune regulation, emphasizing the gut microbiome’s role in shaping host fat-related traits [20]. These studies collectively justify IMF and IMP as robust phenotypic models for dissecting host–microbiome metabolic interactions in poultry.

Emerging evidence highlights the gut microbiome as a major regulator of host metabolism [21]. In giant pandas, metagenomic reconstruction revealed *S. alactolyticus* as a key functional strain that elevates essential amino acid levels in the jejunum through specialized protein metabolism [22]. Similarly, in swine models, methanogenic archaea in pigs with high IMF levels enhance the conversion of acetate to lipids, whereas pigs with low IMF levels are enriched with butyrate-producing Roseburia, which suppresses lipogenic enzyme activity [23]. Lactobacillus reuteri from obese Ningxiang pigs promotes intramuscular fatty acid accumulation by inhibiting PPARα/RXR-mediated β-oxidation in recipient DLY pigs [24]. In poultry, Wen et al. identified distinct cecal microbiota enterotypes in yellow-feathered broilers, with one enterotype exhibiting significantly higher IMF content [25]. In contrast, studies on Guizhou native chickens linking high/low IMF groups found a positive correlation between IMF levels and Alloprevotella/Synergistes abundance, mediated by bile acid metabolism [26]. These insights were facilitated by methodological advances spanning 16S rRNA gene sequencing, shotgun metagenomics, and MAG analysis, with novel technologies like high-fidelity (HiFi) sequencing further enhancing genome recovery precision [27,28].

The chicken cecum, which harbors the densest microbial consortium and is dominated by *Firmicutes* (45–54%) and *Bacteroidota* (23–40%) [29,30] serves as a metabolic reactor in which *Bacteroides* spp. degrade polysaccharides into short-chain fatty acids (SCFAs), such as acetate, propionate, and butyrate [31]. These metabolites modulate nutrient absorption, immune tone, and energy allocation, all of which are closely linked to muscle development [32]. However, two critical knowledge gaps remain: whether the cecal microbiota contributes to differences in IMP/IMF levels among specialized broiler lines, and which microbial taxa and metabolic functions mechanistically underpin these traits.

Here, we employed multi-omics approaches to address these questions in Daheng broilers divergently selected for high and low IMP/IMF contents. Our objectives were to (i) characterize the cecal microbiome structures associated with IMP/IMF phenotypes using 16S rRNA gene sequencing; (ii) validate and refine taxonomic linkages using metagenomics and identify candidate taxa; (iii) reconstruct metabolic pathways and metagenome-assembled genomes (MAGs) to elucidate the microbial functions underlying IMP/IMF variation; and (iv) establish a publicly accessible MAG database for the chicken gastrointestinal microbiome. We hypothesize that the divergent selection for IMP/IMF has shaped distinct cecal microbial communities, and that these microbiota differences contribute directly to the observed variations in IMP/IMF deposition through modulation of specific microbial metabolic functions. This study provides novel evidence supporting microbiome-mediated enhancement of meat quality in precision poultry breeding and offers potential translational strategies for microbiota-guided meat quality optimization.

## 2. Materials and Methods

### 2.1. Experimental Design and Animal Management

Two specialized broiler lines developed by Sichuan Daheng Poultry Breeding Co. (Chengdu, China) were used in this study. The high-IMP/IMF group comprised Line 007 (genotype CCAAMM), which was selected via marker-assisted selection for genes *ADSL*, *GARS-AIRS-GART*, and *A-FABP*. At 90 days of age, this line exhibited high levels of inosine monophosphate (IMP, 3.077 mg/g) and intramuscular fat (IMF, 7.164%) in breast muscle. The Control group consisted of Line S01, which showed lower IMP content (1.374 mg/g) and higher IMF content (9.111%).

Sixty healthy hens (30 per group, 100 days old) were raised in standard cages at Daheng Poultry Breeding Base (Chengdu, China; 30°34′44″ N, 104°4′21″ E). The environmental conditions were maintained at 22 °C and 55–65% relative humidity. Birds had ad libitum access to water and a commercial diet. All procedures complied with the Chinese national standards GB/T 35892-2018 [33] and GB/T 35823-2018 [34] (Ethics approval: L2024029).

### 2.2. Sample Collection and Phenotyping

After 12 h of fasting, cecal contents were aseptically collected within 1–2 h post-slaughter. Samples were immediately flash-frozen in dry ice and stored at −80 °C. Phenotypic traits, including body weight, blood lipid profiles, and immune cell ratios, were measured. Significant intergroup differences (*p* < 0.05) in growth, immunity, and metabolic parameters confirmed the phenotypic divergence between the two groups.

### 2.3. DNA Extraction and Sequencing

#### 2.3.1. 16S rRNA Gene Sequencing

Genomic DNA was extracted from cecal contents using the QIAamp Fast DNA Stool Mini Kit (Qiagen, Hilden, Germany). The V4 hypervariable region of the 16S rRNA gene was amplified using primers 515F and 806R and the amplicons were sequenced on an Illumina NovaSeq 6000 platform (2 × 150 bp; Novogene Co., Tianjin, China). Sequencing libraries met the following quality control criteria: DV200 > 85% and Q30 ≥ 85%.

#### 2.3.2. Metagenomic Sequencing

Twenty representative samples (ten per group) were selected for the shotgun metagenomic sequencing. Genomic DNA was extracted using the QIAamp PowerFecal DNA Kit (Qiagen), fragmented to an average size of 350 bp, and sequencing libraries were prepared using the TruSeq DNA PCR-Free Kit (Thermo Scientific, Waltham, MA, USA). Sequencing was performed using a NovaSeq 6000 platform (PE150, Q30 ≥ 90%).

### 2.4. Bioinformatics Analysis

#### 2.4.1. 16S rRNA Gene Sequencing Data Processing

Raw 16S rRNA gene sequencing reads were processed using QIIME2 (v2023.5) [35]. Demultiplexing and quality filtering were conducted with the Deblur plugin. Feature tables were generated based on 99% operational taxonomic unit (OTU) clustering, and taxonomic assignment was performed using the SILVA v138 reference database. Alpha diversity was assessed using the Shannon, Simpson, and Chao1 indices, whereas beta diversity was evaluated via Bray–Curtis distance and visualized through principal coordinates analysis. Linear discriminant analysis effect size (LEfSe) was applied with a threshold of LDA > 3.0 and a significance level of *p* < 0.05 to identify discriminatory microbial taxa.

#### 2.4.2. Metagenomic Analysis

For metagenomic analysis, raw reads were quality-filtered using Kneaddata (v0.7.4) with host genome filtering (GRCg7b). Taxonomic profiling was performed using Kraken2 (GTDB r207) [36]. MAGs were generated via assembly with MEGAHIT (v1.2.9) [37], followed by binning using MetaBAT2 (v2.15) and quality assessment using CheckM (v1.2.0) to ensure completeness ≥ 50% and contamination ≤ 10%. Dereplication of MAGs was performed using dRep (v3.4.0) at an average nucleotide identity (ANI) threshold of ≥ 95% [38], and functional annotation was performed using GTDB-Tk (r220) [39]. Gene prediction was executed using Prodigal (v2.6.3), and KEGG/GO enrichment analysis was conducted with EggNOG-mapper (v2.1.6).

#### 2.4.3. Chicken MAG Database

Publicly available chicken MAGs (Web of Science; keywords: “chicken MAGs” OR “poultry metagenome”) were integrated with our data. Non-redundant genomes (ANI ≥ 95%) were annotated using GTDB-Tk and visualized using iTOL (v6.7).

### 2.5. Statistical Analysis

Statistical analyses were conducted using R (v4.1.0) to evaluate differences in alpha diversity using the Wilcoxon test, and beta diversity was assessed using PERMANOVA with 999 permutations. *p*-values were adjusted for false discovery rate (FDR) where applicable. For phenotypic traits (e.g., body measurements and slaughter weights), comparisons between high- and low-IMP/IMF groups were performed using independent two-sample *t*-tests after confirming normality (Shapiro–Wilk test) and homogeneity of variance (Levene’s test). Data are presented as mean ± SEM, with statistical significance defined as *p* < 0.05.

## 3. Results

### 3.1. Phenotypic Measurements

Prior to microbial sequencing, we evaluated multiple phenotypic traits in Daheng broilers from high- and low-IMP/IMF groups. Significant inter-group differences (*p* < 0.05) were observed in body slanting length, back width, live body weight, lymphocyte ratio, granulocyte ratio, hemoglobin concentration, hemoglobin content, platelet count, mean platelet volume, platelet distribution width, high-density lipoprotein cholesterol (HDL-C), and triglycerides (TG). These findings (Table 1) indicate divergent growth performance, immune function, and lipid metabolism between groups, justifying further microbiota analyses.

### 3.2. 16S rRNA Gene Sequencing

#### 3.2.1. Sequencing Data Quality

Microbial profiling of the cecal contents from high-IMP/IMF (High, *n* = 30) and low-IMP/IMF (Control, *n* = 30) broilers was conducted. After excluding one sample due to transport loss, 59 samples remained and yielded high-quality data, which were subsequently processed using QIIME2. The sequences were truncated to 357 bp and denoised using the Deblur method, resulting in 3611 unique operational taxonomic units (OTUs). A total of 2,590,076 reads were generated, with an average of 43,899 reads and 863 OTUs per sample (Appendix A). BLASTn alignment revealed that over 99% of features had high-confidence matches (E-value < 1 × 10^−50^; Appendix A). Rarefaction curves reached a saturation plateau (Appendix A), indicating sufficient sequencing depth for robust community analysis.

#### 3.2.2. Taxonomic Composition

At the phylum level (Figure 1), *Firmicutes* and *Bacteroidota* together accounted for over 90% of the microbiota. The High group displayed a relatively balanced distribution between these two phyla (*Firmicutes*: 46.8%; *Bacteroidota*: 43.3%), while the Control group exhibited a higher abundance of *Bacteroidota* (49.8%, Δ + 6.5%). Additionally, *Proteobacteria* was enriched 1.6-fold in the High group (4.2% vs. 2.7%). To depict the most abundant classified taxa, Figure 2 displays the mean relative abundance of the top 20 genera ranked by their overall mean abundance across the entire dataset. The predominant genus was Bacteroides (28.7% vs. 27.5%). A notable proportion of unclassified genera (42–44%) was observed, highlighting the taxonomic gaps in the current avian gut microbiome databases.

#### 3.2.3. Diversity Analysis

Alpha diversity indices revealed significant differences between the High and Control groups (Figure 3). Specifically, species richness, as measured by the Chao1 index, was significantly lower in the High group compared to the Control group (Δ − 12.3%, *p* = 0.018). Phylogenetic diversity, assessed using the Faith_PD index, also showed a significant reduction in the High group (Δ − 9.7%, *p* = 0.027). However, community evenness, evaluated by the Shannon and Simpson indices, did not differ between the two groups. Beta diversity analysis (Figure 4) revealed clear separation between the High and Control groups, with PCoA based on both Bray–Curtis and weighted UniFrac distances explaining over 65% of the total variance. ANOSIM further confirmed that inter-group variation was greater than intra-group variation, with significant differences observed for both Bray–Curtis (R = 0.25, *p* = 0.001) and weighted UniFrac distances (R = 0.09, *p* = 0.004).

#### 3.2.4. Differential Microbial Signatures

LEfSe analysis identified 55 significantly enriched microbial features (LDA > 3, *p* < 0.05; Figure 5). Among them, 34 features were enriched in the high-IMP/IMF group, including multiple Bacteroides taxa (Features 1, 5, and 6), whereas 21 features were enriched in the control group, such as Bacteroides (Feature 4) and unclassified Bacteroidales (Features 11 and 13). Notably, all biomarkers with LDA scores exceeding 4 were affiliated with Bacteroides (Appendix A), underscoring their potential central role in the regulation of IMP/IMF deposition.

### 3.3. Metagenomic Sequencing

#### 3.3.1. Data Quality Control

Metagenomic sequencing of 20 representative samples (10 per group) generated a total of 205.80 GB of high-quality data, comprising 1,372,032,294 sequences, with a Q30 score exceeding 99.9% (Appendix A). A total of 62,443 microbial taxa were identified, with a cumulative sequence count of 47,644,695.

#### 3.3.2. Diversity and LEfSe Analysis

Alpha diversity indices showed no significant differences between the groups (*p* > 0.05; Appendix A). Beta diversity analysis revealed partial overlap in the PCoA plots based on Bray–Curtis and Jaccard distances (Appendix A). However, ANOSIM testing confirmed significant separation between groups (Bray–Curtis: R = 0.18, *p* = 0.013). LEfSe analysis identified 120 group-specific biomarkers (LDA > 3; Figure 6), with 77 enriched species in the High group, including *Gemmiger* (9.0%), *Blautia* (6.4%), and *Merdivivens faecigallinarum* (LDA > 5). The Control group was enriched with 43 species, such as *Coprenecus stercorigallinarum*, and *Barnesiella merdigallinarum* (LDA > 5).

### 3.4. MAG Reconstruction and Functional Annotation

#### 3.4.1. MAG Characteristics

From the 20 representative samples, a total of 882 medium- to high-quality MAGs were successfully reconstructed, meeting the thresholds of ≥50% completeness, ≤10% contamination, and ≥95% ANI. Among these, 287 MAGs were classified as high-quality, exhibiting >90% completeness and <5% contamination. Notably, 50% of the MAGs had completeness ranging from 80% to 100%, and another 50% demonstrated minimal contamination (0–1%) (Appendix A, Appendix A).

#### 3.4.2. Taxonomic Profiling

Taxonomic assignment using GTDB-Tk revealed 220 bacterial species among the reconstructed MAGs (Figure 7). The dominant phyla were *Bacillota_A* (53.2%), *Bacteroidota* (21.4%), and *Bacillota_I* (9.1%). Group-specific taxa were observed, with *Mediterraneibacter* (8.2%) being enriched in the High group and *Alistipes* (6.1%) significantly enriched in the Control group. Additionally, archaeal species such as *Methanobrevibacter_A woesei* were detected, indicating the presence of archaea in the gut microbiota.

#### 3.4.3. Functional Enrichment

KEGG pathway analysis (Figure 8) revealed that the High group was significantly enriched in several pathways, including biosynthesis of cofactors, carbon metabolism, and nucleotide metabolism. Notably, the methane metabolism and starch and sucrose metabolism pathways were more enriched in the High group compared to the Control group (Appendix A). GO enrichment analysis (Figure 9) further identified key functional categories enriched in the High group, including the following: biological process (BP): nucleoside phosphate metabolic process; molecular function (MF): ribonucleoside triphosphate phosphatase activity; and cellular component (CC): organelle envelope.

### 3.5. Chicken Gastrointestinal MAG Database

Integration of 19,628 publicly available MAGs from eight published studies, with the 882 MAGs generated in this study, resulted in a total of 20,510 genomes (Appendix A). After dereplication at 95% ANI, a non-redundant set of 2609 MAGs was obtained, including 52 MAGs derived from the present study. Taxonomic annotation revealed 15 archaeal and 2595 bacterial species (Figure 10). The dominant phyla identified were *Bacillota_A* (41.5%), *Bacteroidota* (15.7%), and *Pseudomonadota* (11.6%). Among the top genera were *Mediterraneibacter* (1.9%), *Alistipes* (1.73%), and *Bacteroides* (1.43%) (Figure 11). This constitutes the largest curated MAG dataset currently available for poultry gut microbiome research.

## 4. Discussion

### 4.1. Overview of Core Findings

This study systematically revealed the structural, functional, and taxonomic differences in the cecal microbiota of Daheng broilers selectively bred for high and low levels of IMP/IMF. After confirming significant phenotypic differences between the two groups in growth performance (e.g., body slant length, pre-slaughter live weight), immunity (lymphocyte/granulocyte ratio), and lipid metabolism (HDL-C, TG), integrated multi-omics analyses demonstrated significant divergence in their cecal microbiota in terms of α/β diversity, taxonomic composition, and metabolic functions. The cross-validation using 16S rRNA gene sequencing, metagenomic sequencing, and MAG analysis robustly supports the conclusion that gut microbial communities play a regulatory role in key meat quality traits.

### 4.2. Microbial Community Structure–Phenotype Association Revealed by 16S rRNA and Metagenomic Sequencing

Although 16S rRNA gene sequencing is cost-effective and technically mature, it requires complementary metagenomic sequencing to achieve higher taxonomic resolution, more accurate species identification, and comprehensive functional annotation [40,41]. In this study, 16S analysis revealed significantly higher Chao1 and Faith_PD indices in the low-IMP/IMF Control group compared to the High group. However, no such statistical differences were detected in the metagenomic data. This discrepancy may arise from methodological limitations; specifically, 16S rRNA gene sequencing is subject to primer bias, resulting in insufficient coverage of certain microbial taxa [42]. Moreover, the presence of a high proportion of unclassified genera (up to 40%) can lead to underestimation of microbial diversity due to its reliance on reference databases. Nevertheless, β-diversity analyses were highly consistent between both methods. ANOSIM tests confirmed that inter-group variation was significantly greater than intra-group variation, strongly supporting an association between cecal microbial community structure and IMP/IMF phenotypes. This finding is consistent with the microbial–host coevolution theory proposed in poultry [43] and parallels previously reported associations between gut microbiota, host metabolism, phenotypic traits in swine [44].

### 4.3. Regulatory Roles of Signature Microorganisms Identified by LEfSe Analysis

As a core functional bacterial group in the chicken gut, *Bacteroides* has unique PULs that can metabolize plant- and host-derived polysaccharides, making it a key producer of SCFAs with significant impacts on host energy metabolism [45,46,47]. LEfSe analysis of 16S rRNA gene sequencing data revealed 55 potential biomarkers between the Control and High groups, with *Bacteroidetes* dominating (all LDA > 4 were *Bacteroides*), indicating ecologically significant strain-level adaptations within this phylum. Recent studies have reported inconsistent findings regarding the microbial taxa associated with IMF content in chickens. For instance, *Synergistes* and *Subdoligranulum* have been reported to be enriched in the gut microbiota of chickens with high IMF levels [26]. In contrast, another study has suggested that a lower abundance of *vadinBE97* correlates with increased IMF deposition [25]. These discrepancies may stem from differences in genetic backgrounds and rearing environments across study populations, which are known to profoundly affect the composition of the gut microbiota and its metabolic interactions with the host.

LEfSe analysis of metagenomic data identified 77 marker species enriched in the High group; unlike 16S results, Bacteroides was not dominant. Instead, 7 *Gemmiger* species and 5 *Blautia* species were enriched, highlighting the superior resolution of metagenomics for species-level identification. *Gemmiger formicilis*, the type species of *Gemmiger* isolated from chicken cecum, ferments various sugars to produce formic, butyric, and lactic acids, thereby influencing host metabolism [48]. *Blautia* spp. are widely distributed in the mammalian gut, with their abundance closely associated with protein and fiber metabolism in human diets [49,50]. *Merdivivens faecigallinarum*, with an LDA score > 5, is the most dominant species in the High group and belongs to the UBA932 family within the phylum Bacteroidota. It was first described by Gilroy et al. [51] and identified through metagenomic assembly in chicken gut microbiota studies. *Alistipes faecavium* is another dominant species in the High group; recent studies suggest that specific *Alistipes* spp. may influence IMF deposition in Guizhou native chickens via the bile acid metabolic pathway [26].

### 4.4. Host–Microbiota Interaction Mechanisms Uncovered by MAG Functional Analysis

IMP/IMF synthesis and degradation metabolism are core indicators of meat quality evaluation and directly influence their deposition in muscle tissues. IMP synthesis occurs via two main pathways: de novo synthesis, using phosphoribosyl pyrophosphate (PRPP) as a precursor, and salvage synthesis, utilizing purine bases [6]. IMF deposition depends on adipocyte proliferation and hypertrophy, along with the synthesis, transport, and degradation of fatty acids [52]. Multiple studies have reported that gut microbiota can influence host lipid metabolism through metabolites such as SCFAs and bile acids [53,54]. *Parabacteroides* and *Bacteroides*, induced by conjugated linoleic acid, along with SCFAs, are closely associated with IMF deposition in pigs [55]. Fecal transplantation from obese Ningxiang pigs demonstrated that *Lactobacillus reuteri* increases IMF content in lean DLY pigs by downregulating the expression of the carnitine transporter SLC22A5 [24]. The rumen microbiota of yak regulates insulin secretion via metabolites such as SCFAs and acetylcholine, upregulates *SREBF1* gene expression through the gut–brain axis, and activates the PPARγ signaling pathway via the gut–muscle axis, thereby promoting the expression of lipogenic genes including *SCD* and *FASN* [56].

In the High group, 122 of the 220 reconstructed bacterial MAGs were detected, with 8.2% assigned to the genus *Mediterraneibacter*, which is known for its metabolic regulatory potential. These MAGs were functionally annotated to KEGG pathways including cofactor biosynthesis, methane metabolism, nucleotide metabolism, and starch and sucrose metabolism, as well as GO biological processes related to nucleotide phosphorolysis.

Cofactors participate in many biochemical reactions, serving as both substrates and products, regulating metabolic networks, signal transduction, and substance transport, thus affecting the physiological functions of microbial cells [57]. Carbohydrate utilization is crucial for chicken growth, and in this study, pathways related to carbohydrate and amino acid metabolism were significantly enriched, consistent with previous studies [58,59]. The succinate pathway for propionate formation primarily occurs in the cecum. *Bacteroidetes* and *Verrucomicrobia* capable of producing propionate have been identified in the poultry gut. These bacteria utilize enzymes such as methylmalonyl-CoA mutase, methylmalonyl-CoA decarboxylase, and methylmalonyl-CoA epimerase [60]. Methanogens are typically abundant in the cecum of adult hens, where they convert CO_2_, methanol, and acetate to methane under anaerobic conditions. Competition with acetogens for hydrogen forces acetogens to use alternative substrates, thereby affecting the production of other organic acids during fermentation [61,62]. This study also successfully reconstructed a *Methanobrevibacter woesei* MAG from the High group, consistent with the metabolic characteristics observed in this group.

### 4.5. Construction and Significance of the Chicken MAG Database

In recent years, significant advances have been made in the characterization of avian gastrointestinal MAGs. However, managing vast and complex metagenomic data faces significant challenges due to computational resource limitations [63]. To address this, lightweight public metagenomic databases have been developed to optimize retrieval efficiency for animal-derived data, such as the broad-scope AnimalMetagenome DB [64] and specialized resources like the Pbac database (Panda gut microbial database) [65]. This study integrated 19,628 MAGs from eight published studies with 882 newly generated MAGs. After dereplication at 95% average nucleotide identity (ANI), a non-redundant chicken gastrointestinal MAG database containing 2609 genomes was constructed. Among these, 52 high-quality genomes were derived from the present experimental populations (high- and low-IMP/IMF groups), including key functionally relevant genera such as *Mediterraneibacter* and *Alistipes*. The phylum-level composition of the 2595 bacterial MAGs—*Bacillota_A* (41.5%), *Bacteroidota* (15.7%), *Pseudomonadota* (11.6%), *Actinomycetota* (7.7%), *Bacillota* (6.1%), and *Bacillota_I* (5.9%)—aligns well with findings by Feng et al. [66].

### 4.6. Limitations and Future Directions

This study rapidly identified differences in microbial community structure using 16S rRNA gene sequencing, while metagenomic analysis enhanced taxonomic resolution to the species level and enabled the identification of marker taxa. MAGs were reconstructed directly from the samples, providing insights into strain-level variations and associated functional pathways. However, the present study does not establish a causal relationship between microbial community differences and variations in IMP/IMF levels, and it lacks quantitative data on key metabolic intermediates such as SCFAs and nucleotides, which limits the mechanistic depth of the analysis. Future research should aim to isolate and culture microbial biomarkers from the high-IMP/IMF group, assess their effects on meat quality through gavage experiments, and integrate cecal metabolomics and host transcriptomics to construct a comprehensive “microbiota–metabolite–host gene” regulatory network.

## 5. Conclusions

In conclusion, our comprehensive analysis of the Daheng broiler cecal microbiome has revealed significant associations between specific microbial taxa and IMP/IMF content, supported by functional enrichment analysis and MAG reconstruction. While our findings indicate potential microbial regulatory mechanisms, further validation is required to establish causality. Future research will focus on isolating and characterizing key microbial strains, conducting oral gavage trials, and integrating multi-omics data to elucidate the precise mechanisms underlying IMP/IMF deposition. This work represents a crucial step towards harnessing microbiome-informed strategies (e.g., microbial biomarker-assisted selection or precision probiotics) to enhance meat quality in commercial poultry breeding.

## Figures and Tables

**Figure 1 animals-15-02337-f001:**
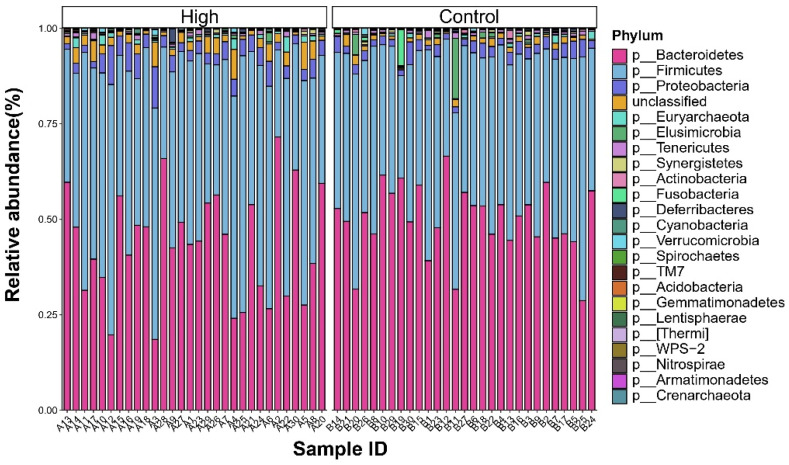
Phylum-level composition of cecal microbiota in high- and low-IMP/IMF broiler groups.

**Figure 2 animals-15-02337-f002:**
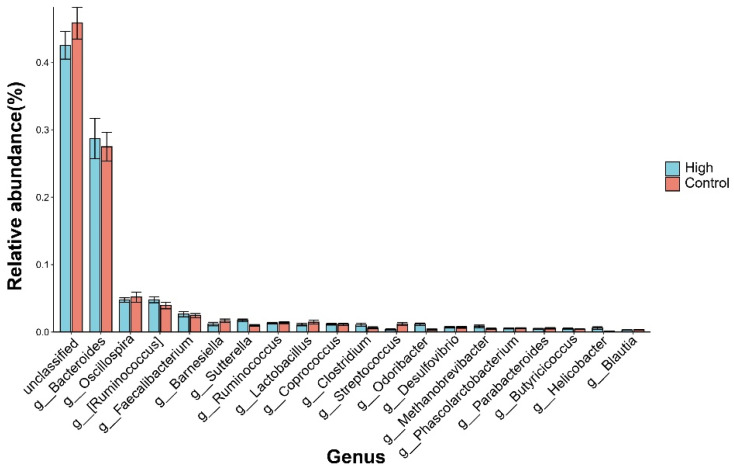
Relative abundance of the predominant bacterial genera in cecal microbiota.

**Figure 3 animals-15-02337-f003:**
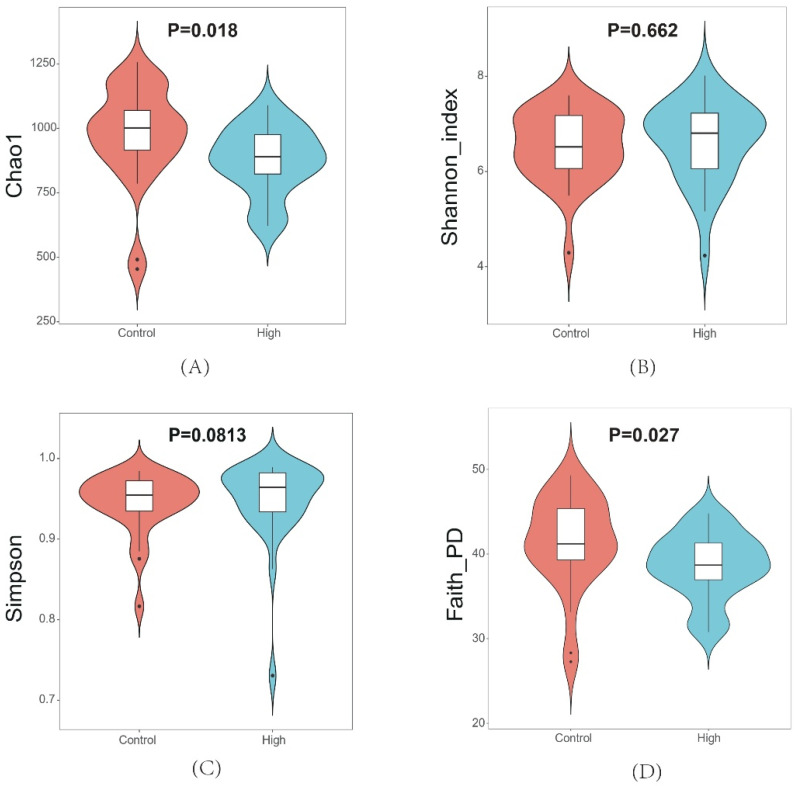
Alpha diversity indices comparing microbial communities. (**A**) Chao 1; (**B**) Shannon; (**C**) Simpson; (**D**) Faith_PD.

**Figure 4 animals-15-02337-f004:**
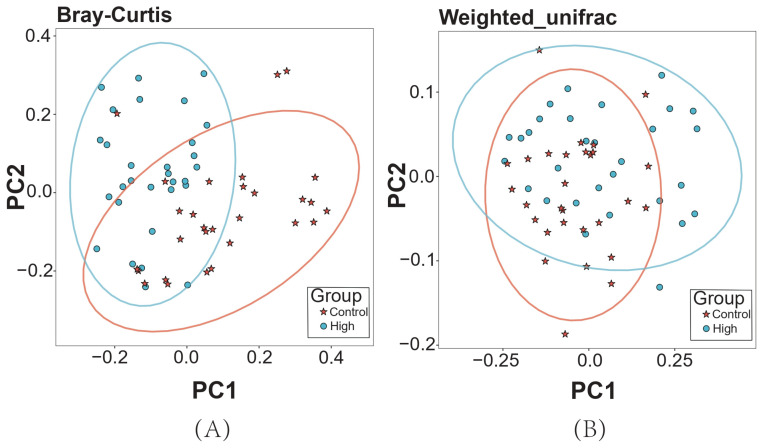
Beta diversity analysis using PCoA plots. (**A**) Bray–Curtis distance. (**B**) Weighted UniFrac distance.

**Figure 5 animals-15-02337-f005:**
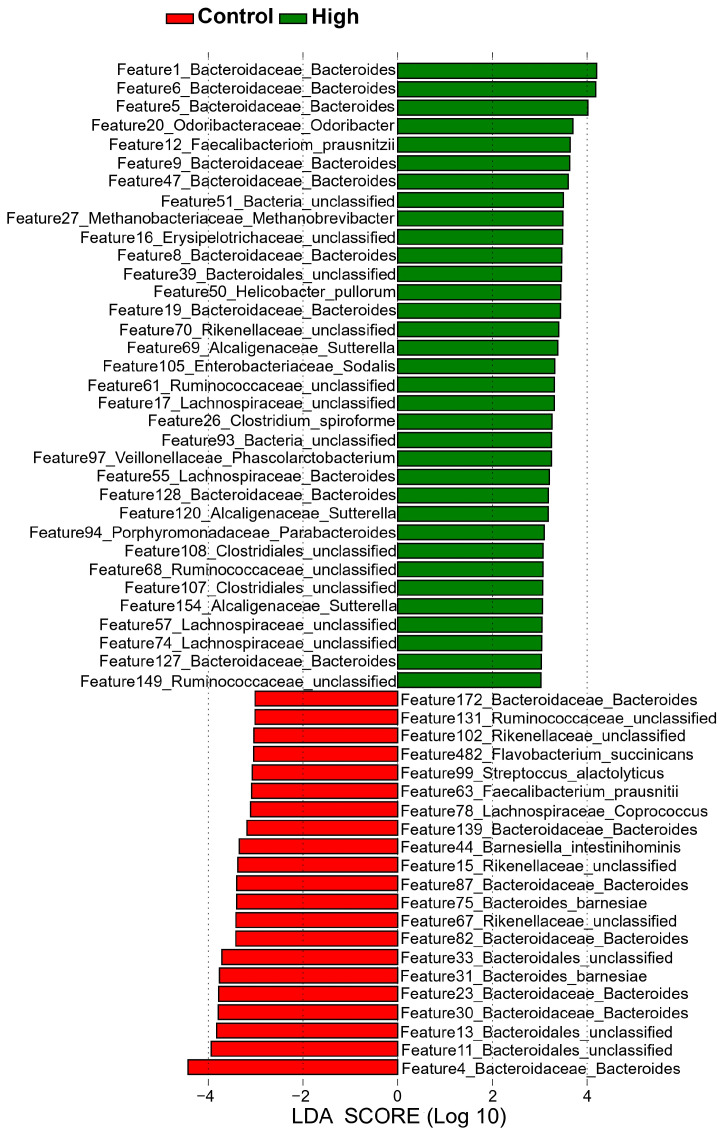
LEfSe identification of differentially enriched taxa. Histogram shows LDA scores for biomarkers (LDA > 3, *p* < 0.05). Red: High group. Green: Control group.

**Figure 6 animals-15-02337-f006:**
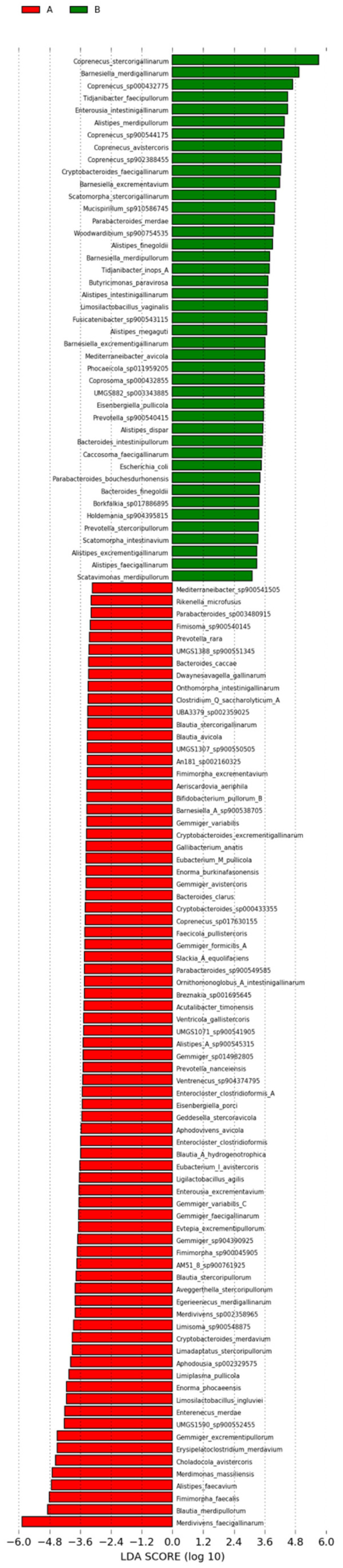
LEfSe cladogram of metagenomic biomarkers (LDA > 3).

**Figure 7 animals-15-02337-f007:**
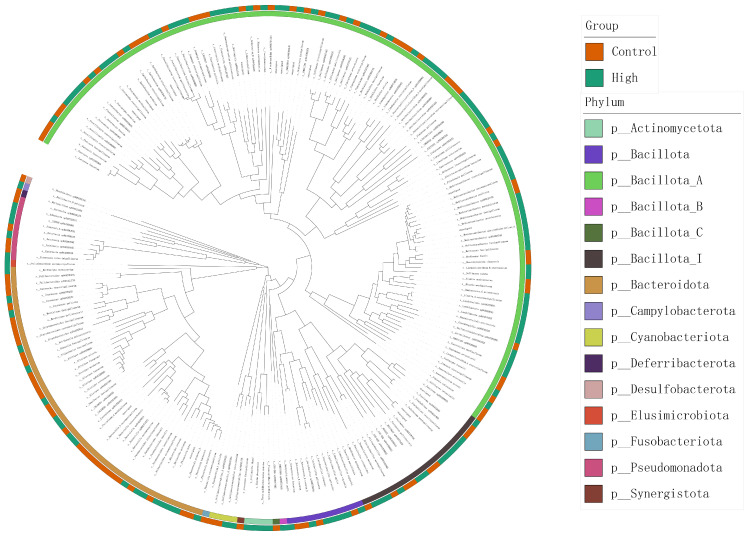
Phylogenetic tree of 220 bacterial MAGs.

**Figure 8 animals-15-02337-f008:**
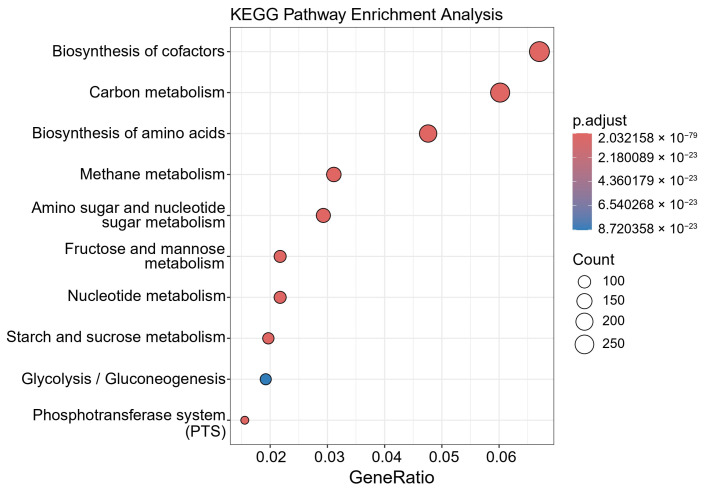
Top 10 KEGG pathway enrichment in High group.

**Figure 9 animals-15-02337-f009:**
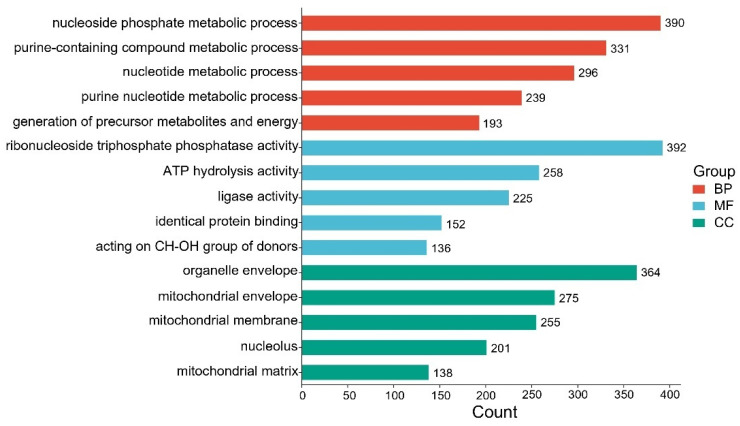
GO enrichment analysis in High group.

**Figure 10 animals-15-02337-f010:**
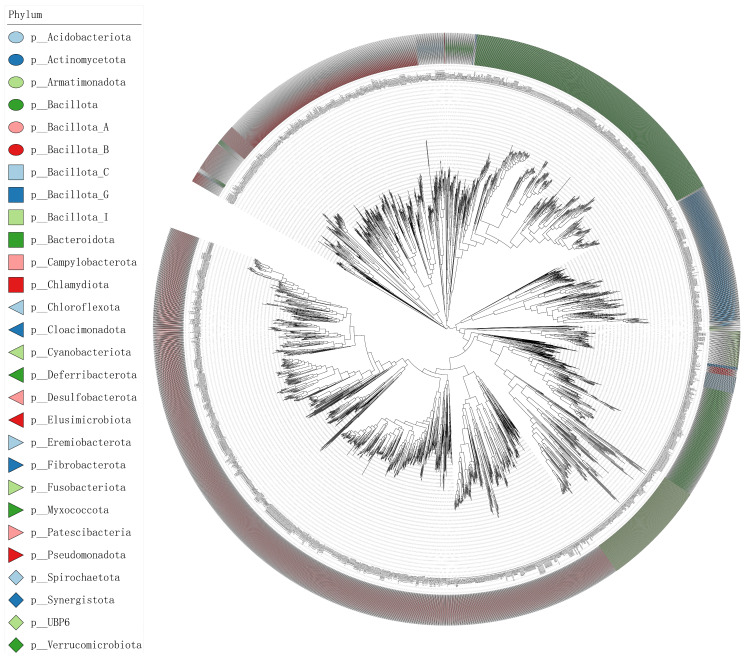
Phylogenetic tree of 2595 bacterial genomes.

**Figure 11 animals-15-02337-f011:**
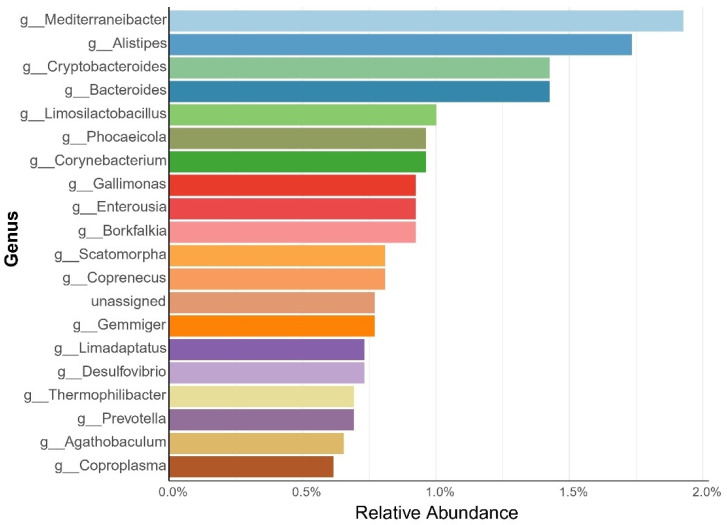
Genus-level abundance in the chicken MAG database, highlighting the top 20 genera comprising 20% of the total.

**Table 1 animals-15-02337-t001:** Phenotypic traits showing significant differences (*p* < 0.05) between high- and low-IMP/IMF broiler groups.

Penotypic Traits	High Group	Control Group	*p* Value
Body slanting length (cm)	19.72 ± 0.17	18.20 ± 0.17	<0.0001
Back width (cm)	7.62 ± 0.08	7.34 ± 0.08	0.018
Slaughter live weight (g)	1251.86 ± 26.83	1080.69 ± 26.83	0.0003
Lymphocyte ratio (%)	77.78 ± 2.41	70.57 ± 2.41	0.046
Granulocyte ratio (%)	13.31 ± 2.09	20.09 ± 2.09	0.032
Hemoglobin concentration (g/L)	106.83 ± 5.97	89.08 ± 5.97	0.048
Hemoglobin content (pg)	46.66 ± 1.70	41.63 ± 1.70	0.048
Platelet count (10^9^/L)	53.75 ± 3.53	66.58 ± 3.53	0.017
Mean platelet volume (fL)	11.95 ± 0.19	11.20 ± 0.19	0.009
Platelet distribution width (%)	19.64 ± 0.72	17.15 ± 0.72	0.023
HDL-C (mmol/L)	1.24 ± 0.06	1.47 ± 0.06	0.018
TG (mmol/L)	0.95 ± 0.10	0.54 ± 0.10	0.016

## Data Availability

Data will be available upon request from the corresponding author.

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
