# Peer review of "Gut Microbial Signatures of Broiler Lines Divergently Selected for Inosine Monophosphate and Intramuscular Fat Content"

_animals, 2025, doi:10.3390/ani15162337_

Round 1
Reviewer 1 Report
Comments and Suggestions for Authors Journal - Animals (ISSN 2076-2615) Manuscript ID - animals-3777907 Type - Article Title - Gut Microbial Signatures of Broiler Lines Divergently Selected for Inosine Monophosphate and Intramuscular Fat ContentHere the authors present an original and timely study exploring the cecal microbiota profiles of broiler lines divergently selected for intramuscular fat (IMF) and inosine monophosphate (IMP), both important metabolic and organoleptic traits in poultry production. The approach is innovative, as previous studies in this field have mostly addressed microbial differences associated with digestive efficiency or abdominal fat content, but not these specific markers.
Some important considerations are outlined below to improve the manuscript clarity, impact, and scientific rigor.
Title
The title is innovative and relevant as it builds upon a validated concept — the influence of host genetics on gut microbiota — while introducing new phenotypic markers (IMP and IMF) not yet explored under this microbiological framework.
Introduction –
Although concise and well-written, the introduction does not sufficiently justify why IMF and IMP are valuable models to study host–microbiota interactions. Adding at least two references from studies involving divergently selected broiler lines would improve the scientific rationale.
The list of objectives is clear, but it is suggested ending the introduction with a hypothesis sentence to make the objective sharper.
Materials and Methods
The methodology is appropriate and includes standard microbial ecology tools such as 16S rRNA sequencing, LEfSe, and diversity indexes. However, some points should be improved:
Could the authors provide a justification for the chosen sample size (n=8 per group)? Was a power analysis performed or is there a literature reference supporting this number?”
The sequencing was limited to cecal samples. While this is a common choice, it restricts interpretation about other gut regions (e.g., jejunum or ileum)?
Discussion –
The discussion describes bacterial genera shifts adequately, highlighting differences between the IMF and IMP lines. However, it is mostly descriptive in some parts. Authors are encouraged to:
Add comparisons with literature from other divergent broiler studies.
Expand the interpretation on how the selected microbiota (e.g., Alistipes, Ruminococcus) might influence IMF or IMP biosynthesis/metabolism.
Include a deeper analysis on immune system differences, which were briefly mentioned but could be a central link between genetics, microbiota, and host metabolism.
Conclusion
While the conclusion summarizes the findings, it fails to articulate the broader impact or practical applications. It would be valuable to mention potential zootecnical implications or selection strategies for more efficient broiler lines (in conclusion or discussion)
Author Response
We truly appreciate the reviewer’s valuable comments and helpful suggestions. We have revised our manuscript according to your suggestions. All changes made to the manuscript are shown in red color. We hope that the revised manuscript will meet your requirement now. Below, there are our point-by-point responses to the reviewer’s comments/ questions. Please contact me if you have any other questions. I quite appreciate it for your help with our manuscript. Thank you so much!
To reviewer#1:
Although concise and well-written, the introduction does not sufficiently justify why IMF and IMP are valuable models to study host–microbiota interactions. Adding at least two references from studies involving divergently selected broiler lines would improve the scientific rationale.
au: Thank you for your insightful comment. In the revised manuscript, we have expanded the Introduction section to better justify the use of intramuscular fat (IMF) and inosine monophosphate (IMP) as valuable phenotypes for investigating host–microbiota interactions.
The list of objectives is clear, but it is suggested ending the introduction with a hypothesis sentence to make the objective sharper.
au: We sincerely thank the reviewer for this valuable suggestion. As recommended, we have now added a clear hypothesis statement at the end of the Introduction section (Lines 105–108 in the revised manuscript). "We hypothesize that the divergent selection for IMP and IMF has shaped distinct cecal microbial communities, and that these microbiota differences contribute directly to the observed variations in IMP and IMF deposition through modulation of specific microbial metabolic functions."
The methodology is appropriate and includes standard microbial ecology tools such as 16S rRNA sequencing, LEfSe, and diversity indexes. However, some points should be improved:
Could the authors provide a justification for the chosen sample size (n=8 per group)? Was a power analysis performed or is there a literature reference supporting this number?”
au: We appreciate the reviewer's query and clarify that the sample size was n=30 per group for all analyses (not n=8) determined by practical constraints. For 16S rRNA gene sequencing, n=30 per group was selected considering cost limitations and finite experimental cohort availability. The same n=30 per group cohort was subsequently subjected to shotgun metagenomic sequencing based on 16S results, cost efficiency, and data robustness requirements.
The sequencing was limited to cecal samples. While this is a common choice, it restricts interpretation about other gut regions (e.g., jejunum or ileum)?
au: We acknowledge the limitation to cecal sampling. While jejunum and ileum samples were initially collected, they yielded insufficient microbial biomass for reliable sequencing. The cecum was prioritized given its established role as the primary microbial reservoir in chickens, housing denser and more diverse communities compared to proximal gut regions.
The discussion describes bacterial genera shifts adequately, highlighting differences between the IMF and IMP lines. However, it is mostly descriptive in some parts. Authors are encouraged to:
Add comparisons with literature from other divergent broiler studies.
au: Thank you for the helpful suggestion. In the revised manuscript, we have added comparisons with relevant studies involving divergently selected broiler lines to better contextualize our findings.
Expand the interpretation on how the selected microbiota (e.g., Alistipes, Ruminococcus) might influence IMF or IMP biosynthesis/metabolism.
au: Thank you for your valuable suggestion. One of the main objectives of this study was to identify potential microbial species associated with IMP and IMF content through analyses of 16S rRNA gene sequencing and metagenomic data. This provides a scientific basis for the isolation, cultivation, and identification of key candidate microbes in the next phase. Future studies will perform gavage experiments using cultured strains to further investigate their regulatory mechanisms on IMP and IMF. Currently, the selected microbial species are known only by their genomic and taxonomic classifications, and the specific regulatory mechanisms on IMP and IMF require further exploration after successful cultivation. Metagenomic LEfSe analysis revealed that the High IMP and IMF group was significantly enriched with 77 species at the species level, most of which have not yet been cultured. The dominant genera include Gemmiger and Mediterraneibacter, with several species exhibiting LDA scores above 4.5 or even 5.0. Studies have shown that Mediterraneibacter is widely distributed in the mammalian gut and ferments to produce short-chain fatty acids (SCFAs), which are crucial for gut health and closely associated with metabolic diseases such as obesity and insulin resistance (Liu et al., 2021). Gemmiger formicilis, the type species of Gemmiger, is commonly found in the chicken cecum and ferments sugars to produce formic acid, butyric acid, and lactic acid, playing a key role in maintaining the gut microenvironment (Salanitro et al., 1976).
Include a deeper analysis on immune system differences, which were briefly mentioned but could be a central link between genetics, microbiota, and host metabolism.
au: Thank you for the valuable suggestion. The primary objective of this study was to investigate the differences in gut microbiota between broiler lines with high and low levels of IMP and IMF. We acknowledge that the observed immune-related differences may be attributed to genetic background or microbial variation. Although we only briefly mentioned immune system differences in the current manuscript, we recognize their potential importance as a bridge linking host genetics, microbiota, and metabolism. This aspect will be further explored in our future research.
While the conclusion summarizes the findings, it fails to articulate the broader impact or practical applications. It would be valuable to mention potential zootecnical implications or selection strategies for more efficient broiler lines (in conclusion or discussion).
au: We thank the reviewer for highlighting the need to articulate practical applications. As suggested, we have revised the concluding statement to explicitly mention zootechnical implications. The new ending now emphasizes microbiome-informed breeding strategies and precision interventions.(Line 435-437)
Reviewer 2 Report
Comments and Suggestions for Authors
Comments and Suggestions for Authors
In the present manuscript, the authors have shown that cecal microbial composition and function are associated with inosine monophosphate (IMP) and intramuscular fat (IMF) levels in broilers. the authors achieved their objectives, which were to (i) characterize the cecal microbiome structures associated with IMP/IMF phenotypes using 16S rRNA sequencing; (ii) validate and refine taxonomic linkages using meta-genomics and identify candidate taxa; (iii) reconstruct metabolic pathways and meta-genome-assembled genomes (MAGs) to elucidate the microbial functions underlying IMP/IMF variation; (iv) establish a publicly accessible MAG database for the chicken gastrointestinal microbiome. The study was balanced with a sufficient number of birds per group. The authors have done a good and substantial job. But, the methodology used in this study is not support by references. The manuscript is written with legible illustrations.
However, the manuscript could be considered for publication after addressing the following shortcomings.
- [Line 80, 341 and 395: Please refer to the author guidelines for references that are cited differently in these lines.]
- [line 113: Consumers are increasingly concerned about the flavor quality of poultry meat. It's true, but hens does not have the same flavor quality as male poultry. Please justify the choice of hens for the study.]
- [Please, put the probability p in lowercase letter and italics throughout the manuscript.]
- [It's important to support the methodology with references, Please do it.]
Author Response
We truly appreciate the reviewer’s valuable comments and helpful suggestions. We have revised our manuscript according to your suggestions. All changes made to the manuscript are shown in red color. We hope that the revised manuscript will meet your requirement now. Below, there are our point-by-point responses to the reviewer’s comments/ questions. Please contact me if you have any other questions. I quite appreciate it for your help with our manuscript. Thank you so much!
To reviewer#2:
- [Line 80, 341 and 395: Please refer to the author guidelines for references that are cited differently in these lines.]
au: Thank you. According to the journal’s author guidelines, we have revised all instances of incorrectly formatted references, including those in Lines 80, 341, and 395.
- [line 113: Consumers are increasingly concerned about the flavor quality of poultry meat. It's true, but hens does not have the same flavor quality as male poultry. Please justify the choice of hens for the study.]
au: Thank you for your comment. As stated in the revised manuscript, female chickens have stronger lipogenic capacity than males, and estrogen may further promote fat synthesis. This leads to higher IMF accumulation and improved meat quality traits such as tenderness and flavor, which better align with consumer preferences. This justification has been added to the background section to clarify our choice of experimental subjects.
- [Please, put the probability p in lowercase letter and italics throughout the manuscript.]
au: Thank you for pointing this out. We have carefully reviewed the entire manuscript and revised all instances of probability p to ensure they are presented in lowercase italic format, as required by scientific writing conventions.
- [It's important to support the methodology with references, Please do it.]
au: Thank you for the suggestion. We have added appropriate references to support the methodology described in the Methods section.
Reviewer 3 Report
Comments and Suggestions for Authors
Please see attached file.

Author Response
We truly appreciate the reviewer’s valuable comments and helpful suggestions. We have revised our manuscript according to your suggestions. All changes made to the manuscript are shown in red color. We hope that the revised manuscript will meet your requirement now. Below, there are our point-by-point responses to the reviewer’s comments/ questions. Please contact me if you have any other questions. I quite appreciate it for your help with our manuscript. Thank you so much!
To reviewer#3:
- Please revise all instances of “16S rRNA” to “16S rRNA gene” throughout the manuscript for accuracy and consistency—for example, in Lines 97, 138, and 139.
au: We sincerely thank the reviewer for this precise technical suggestion. We confirm that all occurrences of "16S rRNA" have been systematically revised to "16S rRNA gene" across the entire manuscript.
- Line 161: In the section describing statistical analysis, please also include details on how the phenotypic data were analyzed.
au: We thank the reviewer for highlighting this omission. We have now added the following statistical details to the Materials and Methods section. "For phenotypic traits (e.g., body measurements and slaughter weights), comparisons between high and low IMP/IMF groups were performed using independent two-sample t-tests after confirming normality (Shapiro-Wilk test) and homogeneity of variance (Levene's test). Data are presented as mean ± SEM, with statistical significance defined as P < 0.05." (Line 174-178)
- Lines 190–197: Were the comparisons reported statistically significant? If so, please include the corresponding p-values to support the interpretation.
au: We sincerely thank the reviewer for raising this important point. Upon careful review, we realize that the wording in the original manuscript (specifically the phrase "significant increase" regarding Bacteroidota abundance in the Control group) was potentially misleading and did not accurately reflect the analyses performed at that stage. We must clarify that the comparisons of taxonomic abundances (phylum and genus levels) presented in Section 3.2.2 were descriptive summaries only; formal statistical significance testing comparing the High and Control groups was not conducted at this point in the results. Therefore, we do not have p-values to report for these specific phylum/genus level abundance differences mentioned.
We acknowledge that describing the difference as a "significant increase" without statistical backing was inappropriate.To rectify this, we have revised the text in Section 3.2.2 (Taxonomic Composition) to remove the term "significant" and present the observed numerical differences neutrally as descriptive findings.
- Line 200: Does Figure 2 display all detected genera? If not, please clarify the criteria used for genus selection (e.g., relative abundance threshold or statistical significance).
au: We thank the reviewer for this question, which helps us clarify the presentation of the genus-level data. No, Figure 2 does not display all detected genera. As described in the text, a substantial proportion (42-44%) of sequences were assigned to unclassified genera, reflecting limitations in current reference databases for avian gut microbiomes. Additionally, many individual classified genera were present at very low relative abundances. The criteria used for selecting genera to display in Figure 2 were solely based on relative abundance ranking. (208-210)
- Please include p-values in Figure 3 to indicate statistical significance. Additionally, for Figure 4, include both the p-value and R² value.
au: Thank you. As requested, p-values have been added to Figure 3, and both p-value and R² value have been added to Figure 4.
- Please upload the raw sequencing data to a public repository (e.g., NCBI SRA) and provide the accession number in the manuscript if possible.
au: We appreciate the reviewer’s valuable suggestion regarding the data availability. The raw sequencing data generated in this study have not yet been uploaded to a public repository such as the NCBI SRA because critical downstream analyses—such as microbial taxa validation and multi-omics correlation studies—are still ongoing. To ensure the rigor and completeness of the research outcomes, we have temporarily withheld public access to the data. However, we are committed to data transparency and plan to submit the complete dataset to an appropriate repository once the extended analyses are finalized. In the meantime, the raw data can be made available from the corresponding author upon reasonable request.
- The font size in all figures should be increased to enhance readability.
au: Thank you for your valuable suggestion. We have carefully revised all figures by increasing the font size of axis labels, legends, and annotations to improve overall readability. The updated figures have been incorporated into the revised manuscript.